# DECODING READING GOALS FROM EYE MOVEMENTS

## ABSTRACT

Readers can have different goals with respect to the text they are reading. Can these goals be decoded from the pattern of their eye movements over the text? In this work, we examine for the first time whether it is possible to decode two types of reading goals that are common in daily life: *information seeking* and *ordinary reading*. Using large scale eye-tracking data, we apply to this task a wide range of state-of-the-art models for eye movements and text that cover different architectural and data representation strategies, and further introduce a new model ensemble. We systematically evaluate these models at three levels of generalization: new textual item, new participant, and the combination of both. We find that eye movements contain highly valuable signals for this task. We further perform an error analysis which builds on prior empirical findings on differences between ordinary reading and information seeking and leverages rich textual annotations. This analysis reveals key properties of textual items and participant eye movements that contribute to the difficulty of the task.[1]

## 1 INTRODUCTION

Reading is a ubiquitously practiced skill that is indispensable for successful participation in modern society. When reading, our eyes move over the text in a saccadic fashion, where there are periods of time in which the gaze is stable at a specific location, called *fixations*, and rapid transitions between fixations called *saccades* (Rayner, 1998). This sequence of fixations and saccades is generally hypothesized to contain rich information about how readers interact with text. Automatic decoding of such information is currently a growing area of research (e.g. Reich et al., 2022a; Shubi et al., 2024).

In daily life, a reader may have one or several *goals* that they pursue with respect to the text. For example, they may read the text closely or skim it to obtain the gist of the text's content, they may proofread it, or they may be seeking specific information of interest. Each such goal can have a profound impact on online linguistic processing and on the corresponding eye movement behavior while reading. Despite the many reading goals readers pursue in everyday life, research on eye movements in cognitive science as well as work that integrated eye movements data in Natural Language Processing (NLP) and machine learning (ML) have primarily focused on one reading regime, which can be referred to as *ordinary reading*. In this regime, the reader's goal is typically general comprehension of the text. Nearly all broad coverage datasets used in research on eye movements in reading, such as Dundee (Kennedy et al., 2003), MECO (Siegelman et al., 2022) and CELER (Berzak et al., 2022) were collected in this regime. Other forms of reading, although widely acknowledged (Radach & Kennedy, 2004), received much less attention and remain understudied.

In this work, we go beyond ordinary reading and ask whether broad reading goals can be reliably decoded from the pattern of the reader's eye movements over the text. We focus on the distinction between ordinary reading and *information seeking*, a highly common reading regime in everyday life, where the reader is interested in obtaining specific information from the text. Prior work suggests that on average across participants and texts, there are substantial differences in eye movement patterns between these two reading regimes (Hahn & Keller, 2023; Shubi & Berzak, 2023). However, it is currently unknown whether there is sufficient signal in eye movements for automatic decoding of the reading goal given eye movements of a single participant over a single textual item. Furthermore, little is known about the factors that contribute to the difficulty of this task.

---

[1]Code is available at the following anonymous link: `https://anonymous.4open.science/r/Decoding-Reading-Goals-from-Eye-Movements/`. Data will be made publicly available.

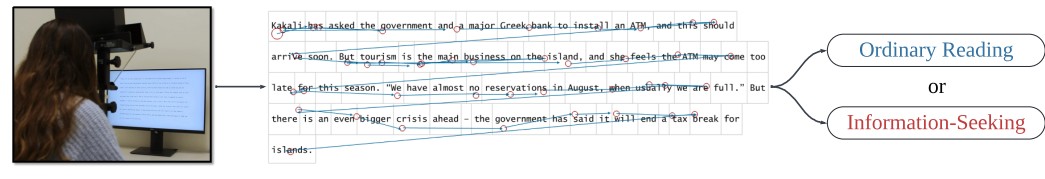

Single Participant $S$      Eye movement recording for a single passage $P$      Participant's Reading Goal

Figure 1: Proposed task: decoding whether a reader is seeking specific information or reading for general comprehension, given their eye movements over a single passage. In the eye movements image, circles represent fixations, and lines represent saccades. Bounding boxes mark word Interest Areas (fixations within the box are assigned to the respective word).

In this work, we address this gap by conducting a series of experiments on reading goal decoding. We use a range of recent models that implement different strategies for representing eye movements and combining them with text, and perform systematic evaluations of their individual and ensemble generalization abilities. We demonstrate that it is indeed possible to perform this task with relatively high accuracy rates. We further perform a systematic error analysis which leverages rich data annotations to reveal key axes of variation that contribute to task difficulty.

The main contributions of this work are the following:

- **Task**: We introduce a new decoding task: given eye movements from a single participant over a passage, predict whether they engaged in ordinary reading or in information seeking.
- **Modeling and Evaluation**: We adapt and apply to this task 10 different state-of-the-art predictive models for eye movements in reading. We further introduce an ensemble model which outperforms the best single model. We characterize the generalization ability of all the models across textual items and participants.
- **Error Analysis**: We use statistical modeling and detailed textual annotations to perform an error analysis which reveals interpretable axes of variation in item and participant-related properties of trials that are indicative of classification difficulty. This analysis provides new insights on the nature of the task.

## 2 TASK

We address the task of predicting whether a reader is engaged in ordinary reading for comprehension or in seeking specific information, based on their eye movements over the text. Let $S$ be a participant, $P$ a textual passage, and $E_P^S$ the recording of the participant's eye movements over the passage. Given a ground-truth mapping $\mathcal{C}(S, P) \rightarrow \{\text{Information Seeking}, \text{Ordinary Reading}\}$, we aim to approximate $\mathcal{C}$ with a classifier $h$:

$$h : (E_S^P, P) \rightarrow \{\text{Information Seeking}, \text{Ordinary Reading}\}$$

Where the passage $P$ is an optional input, such that the classifier can be provided only with the eye movement data $E_S^P$ or with both the eye movements and the underlying text $P$. We assume that the participant has not read the paragraph previously.

The information seeking regime is operationalized by presenting the participant with a question $Q$ prior to reading the passage. This question prompts the participant to seek specific information in the text. We assume that the classifier does not receive the question nor any information on the participant. Figure 1 presents the task schematically.

## 3 MODELING

Eye movements during reading present a highly challenging case of temporally *and* spatially aligned multimodal data, where fixations are temporal and correspond to specific words in the text. We use a wide range of state-of-the-art architectures for processing this data. While the models were originally applied to other prediction tasks, such as predicting participant's reading comprehension and

various participant properties, they offer general purpose mechanisms for processing eye movement data during reading that can be applied to other tasks, and cover different architectural and data representation strategies. The models can be broadly divided along three primary axes, the modalities used (eye movements-only, or eye movements and text), the granularity at which eye movement information is represented (global averages across the text, single word, or a single fixation), and for the multimodal approaches, the nature of the text representations and strategy for combining them with eye movements. These models are directly applicable, in some cases with small adaptations, to the task of reading goal prediction. See Figures 5 and 6 in Appendix A for model diagrams.

### 3.1 Eye Movements-only Models

These models use only eye movement information, without taking into account the text. Such models are valuable in common scenarios where the underlying text for the eye movement recording is not available. It is also the go-to approach when the eye-tracking calibration is of low quality, leading to imprecise information on the location of fixations with respect to the text. This is a highly common situation, especially with web-based eye-tracking and lower grade eye-tracking devices. Beyond practical considerations, the eye movements-only approach allows assessing the added value of textual information for our task.

#### Global Representation

**Logistic Regression** A logistic regression model with global eye movement measures from Mézière et al. (2023). The measures include averages of word reading times, single fixation duration, forward saccade length, the rate of regressions (saccades that go backwards), and skips (words that were not fixated) during first pass reading (i.e. before proceeding to the right of the word). All the features are standard measures from the psycholinguistic literature.

#### Fixation-Based Representation

**BEyeLSTM - No Text** is our adaptation of the BEyeLSTM model (Reich et al., 2022a) described below, which uses only eye movements input without the underlying text. This model is more expressive compared to the logistic regression model, in that rather than averaging eye movement measures for the entire trial, it explicitly represents each fixation in the eye movement trajectory. The final hidden state of an LSTM (see below), combined with global eye movement features, is then used for classification.

### 3.2 Eye Movements and Text Models

We examine a number of recent multimodal models that combine eye movements with textual information. The models encode textual information in two ways. The first is using contextual word embedding representations commonly used in NLP. The second is via linguistic word property features, including word length, word frequency and surprisal (Hale, 2001; Levy, 2008), which are motivated by their ubiquitous effects on reading times (Rayner et al., 2004; Kliegl et al., 2004; Rayner et al., 2011, among others).

The models implement three primary strategies for combining the two modalities at progressively later stages of processing: (i) in the model input, (ii) merging them within intermediate model representations, or (iii) with architectures that fuse the modalities using cross attention mechanisms after each modality has been processed separately. Furthermore, since eye movements in reading are both temporally and spatially aligned with the underlying text, the models can be categorized based on how they capture this alignment: (i) by aggregating eye movements for each *word*, thereby focusing on spatial alignment; or (ii) by aggregating eye movement information for each individual *fixation*, which explicitly encodes both spatial and temporal correspondences between eye movements and text.

#### Word-based Representations

**RoBERTa-Eye-W(ords)** (Shubi et al., 2024) is a RoBERTa transformer model (Liu et al., 2019) augmented with eye movements. This model concatenates word embeddings and word-level eye movement features in the model input.

**MAG-Eye** (Shubi et al., 2024) Integrates word-level eye movement features into a transformer-based language model by injecting them into intermediate word representations using a Multimodal Adaptation Gate (MAG) architecture (Rahman et al., 2020). The text is aligned with eye movements by duplicating each word-level eye movement feature for every sub-word token.

FIXATION-BASED REPRESENTATIONS

**PLM-AS** (Yang & Hollenstein, 2023) This model represents the eye movements sequence by re-ordering contextual word embeddings according to the order of the fixations over the text. This reordered sequence is processed through a Recurrent Neural Network (RNN), whose final hidden layer is used for classification.

**Haller RNN** (Haller et al., 2022) This model is similar to PLM-AS in that it receives word embeddings in the order of the fixations. Differently from PLM-AS, each word embedding is further concatenated with eye movement features.

**RoBERTa-Eye-F(ixations)** (Shubi et al., 2024) uses the same architecture as RoBERTa-Eye-W, but represents fixations rather than words. Each fixation input consists of a concatenation of the word embedding and eye movement features associated with the fixation.

**BEyeLSTM** (Reich et al., 2022a) represents both the fixation sequence and textual features, combining LSTMs (Hochreiter & Schmidhuber, 1997) and global features through a linear layer.

**Eyettention** (Deng et al., 2023) is a model that consists of a RoBERTa word sequence encoder and an LSTM-based fixation sequence encoder. It uses a cross-attention mechanism to align the input sequences. We use the adaptation of this model by Shubi et al. (2024) for binary classification.

**PostFusion-Eye** (Shubi et al., 2024) is a model that consists of a RoBERTa word sequence encoder and a 1-D convolution-based fixation sequence encoder. It then uses cross-attention to query the word representations using the eye-movement representations, followed by concatenation and projection into a shared latent space.

### 3.3 BASELINES

When examining the utility of eye movements for a prediction task, it is important to benchmark models against simpler approaches that do not require eye movement information (Shubi et al., 2024). Here, we introduce two such baselines:

**Majority Class** Assigns the label of the majority class in the training set to all the trials in the test set. Since our data is balanced (see below), this baseline is equivalent to random guessing.

**Reading Time (per word)** Total reading time per word, computed by dividing the total reading time of the paragraph by the number of words in the paragraph. This behavioral baseline does not require eye-tracking and is motivated by the analyses of Malmaud et al. (2020), Hahn & Keller (2023) and Shubi & Berzak (2023), which indicate that on average, reading is faster in information seeking compared to ordinary reading.

## 4 EXPERIMENTAL SETUP

### 4.1 DATA

Addressing the proposed task is made possible by OneStop Eye Movements (Malmaud et al., 2020), the first dataset that contains broad coverage eye-tracking data in both ordinary reading and information seeking regimes. The textual materials of OneStop are taken from OneStopQA (Berzak et al., 2020), a multiple-choice reading comprehension dataset that comprises 30 Guardian articles from the OneStopEnglish corpus (Vajjala & Lučić, 2018). Each article is available in the original (Advanced) and simplified (Elementary) versions. Each paragraph has three multiple choice reading comprehension questions that can be answered based on any of the two paragraph difficulty level versions. Each question is paired with a manually annotated textual span, called the *critical span*, which contains the vital information for answering the question. The mean paragraph length

is 109 words (min: 50; max: 165; std: 28). An example of a OneStopQA paragraph along with one question and its critical span annotation is provided in Table 2 in Appendix B.

Eye movements data for OneStopQA were collected in-lab from 360 adult native English speakers using an EyeLink 1000+ eye tracker (SR Research), a state-of-the-art eyetracking device for reading research. Each participant read a batch of 10 articles (54 paragraphs) paragraph by paragraph and answered a comprehension question about each paragraph. Each paragraph was shown to a participant either in the Advanced or Elementary version, along with one of the three possible questions. The experiment has two between-subjects reading goal tasks: information seeking and ordinary reading. In the information seeking task, participants were presented with the question (without the answers) prior to reading the paragraph. In the ordinary reading task, they did not receive the question prior to reading the paragraph. In both tasks, after having read the paragraph, participants proceeded to answer the question on a new screen, without the ability to return to the paragraph.

In sum, each participant read 54 paragraphs, all of which are either in information seeking or in ordinary reading. Each paragraph was read by 120 participants: 60 in ordinary reading and 60 in information seeking (split equally between the Advanced and Elementary versions of the paragraph). Overall, the data consists of 19,438 trials, where a trial is a recording of eye movements from a single participant over a single paragraph. The data is balanced, with 9,718 trials in ordinary reading and 9,720 in information seeking. Figure 7 in Appendix B shows example trials for each of the two reading regimes. Additional data statistics are described in Appendix B.

## 4.2 MODEL TRAINING AND EVALUATION PROTOCOL

We adopt the training and evaluation procedure of Shubi et al. (2024) for prediction of reading comprehension from eye movements, with several modifications for our task. Similarly to Shubi et al. (2024), we use 10-fold cross validation, addressing three levels of model generalization:

- New Item (paragraph): eye-tracking data is available during training for the participant but not for the paragraph.
- New Participant: eye-tracking data is available during training for the paragraph, but not for the participant.
- New Item & Participant: Neither the participant nor the paragraph are in the training data.

We further report aggregated results across all three regimes.

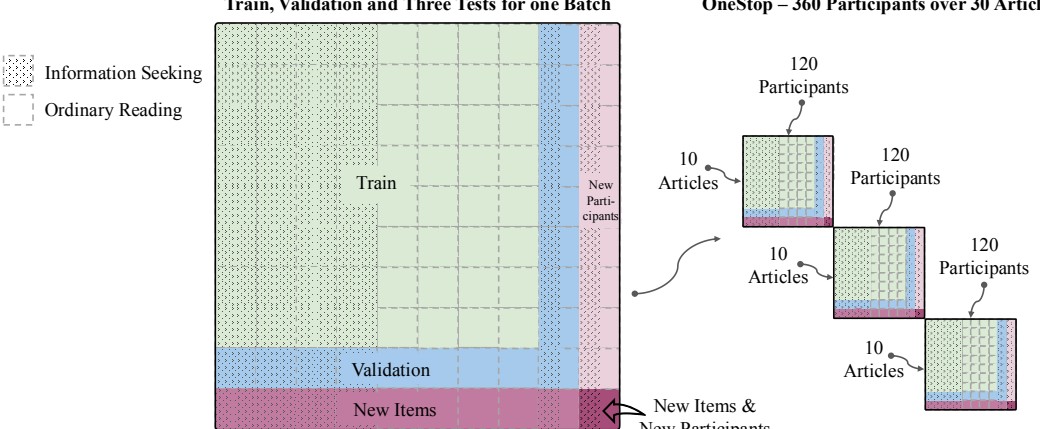

Figure 2: A schematic depiction of one of the 10 splits into train, validation, and the three test sets for one batch of 10 OneStopQA articles and 120 participants. Dashed lines denote information seeking trials. The full data split consists of the union of three such splits. Adapted from (Shubi et al., 2024).

The data is split separately for each batch of 10 articles and the 120 participants who read the batch. The three batch splits are then combined to form the full split of the dataset. Paragraphs are

allocated to the train, validation and test portions of each batch split at the *article level*, such that all the paragraphs of each article appear in the same portion of the split. This ensures that items in the test set are unrelated in content to items in training and validation. Each split further guarantees an equal number of participants from each OneStopQA batch in each data portion.

Shubi et al. (2024) predicted reading comprehension separately in information seeking and ordinary reading, and thus defined separate data splits for the information seeking and ordinary reading trials. They further stratified items by the selected answers to the reading comprehension questions. As these steps are not relevant for our task, we use the union of the information seeking and ordinary reading trials in each part of each split, and stratify trials by reading goal rather than by reading comprehension answers.

Each data split contains approximately 64% of the trials in the training set, 17% in the validation set and 19% in the test sets (9% New Item, 9% New Participant and 1% New Item & Participant). Aggregated across the 10 splits, approximately 90% of the trials in the dataset appear in each of the New Participant and New Item evaluation regimes, and 10% in the New Item & Participant regime. Figure 2 presents this breakdown for one batch split.

We perform hyperparameter optimization and model selection separately for each split. We assume that at test time, the evaluation regime of the trial is *unknown*. Model selection is therefore based on the entire validation set of the split. All neural network-based models were trained using the PyTorch Lighting (Falcon & team, 2024) library on NVIDIA A100-40GB and L40S-48GB GPUs. Further details regarding the training procedure, including the full hyperparameter search space for all the models are provided in Appendix C.

### 4.3 STATISTICAL TESTS FOR MODEL PERFORMANCE COMPARISONS

The samples in the OneStop dataset are not i.i.d; each item is read by multiple participants, and each participant reads multiple items. To account for these dependencies when performing statistical tests, we fit linear mixed-effects models with maximal random effects for items and participants (Barr et al., 2013). To this end, we use the MixedModels package in Julia (Bates et al., 2024).

## 5 RESULTS

Test set accuracy results are presented in Table 1. In line with prior findings regarding faster reading in information seeking compared to ordinary reading (Malmaud et al., 2020; Hahn & Keller, 2023; Shubi & Berzak, 2023), the Reading Time baseline yields above chance accuracies ($p < 0.01$ in all evaluation regimes), thus providing a strong benchmark for the eyetracking-based models. Among the 10 examined models, RoBERTa-Eye-F achieves the highest accuracy in the New Item, New Item & Participant and All evaluations, while RoBERTa-Eye-W slightly outperforms it in the New Participant regime. Differently from the New Item and the New Participant regimes, where multiple models outperform the Reading Time baseline, in the most challenging New Item & Participant regime, the improvements over this baseline are not statistically significant.

In addition to RoBERTa-Eye-F, three other fixation based models, PostFusion-Eye and the two BEyeLSTM models, excel in the New Item regime. One possible explanation for this outcome is better generalization of fixation based models to new items compared to new participants. Note however, that due to the between-subjects design where all the training and test examples for a given participant have the same label, it could alternatively reflect, at least in part, an ability of fixation based models to learn participant specific reading behavior without explicit information on the participant (i.e. identify the participant), an ability that is not directly pertinent to the task at hand. The current experimental setup makes it challenging to adjudicate between these two possibilities. In either case, it is highly non-trivial that models are able to generalize from prior participant data to new items. In Appendix D, we report the Receiver Operating Characteristic (ROC) curves across the ten cross-validation splits and their corresponding Area Under the ROC Curve (AUROC) scores (mean and standard deviation) (Bradley, 1997) in Figure 8. Validation set accuracies are reported in Table 3. The outcomes of these evaluations are consistent with the test results reported in Table 1.

Table 1: Test accuracy results aggregated across 10 cross-validation splits. 'E' stands for word embeddings, 'LF' for linguistic word properties such as word length, frequency and surprisal, and 'Fix' for fixations. Model performance is compared to the Reading Time baseline using a linear mixed effects model. In R notation: $is\_correct \sim model + (model \mid participant) + (model \mid paragraph)$. Significant gains over this baseline are marked with '*' $p < 0.05$, '**' $p < 0.01$ and '***' $p < 0.001$ in superscript, and significant drops compared to the best model in each regime are marked in subscript with '+'.

| Model | Gaze Rep. | Text Rep. | New Item | New Participant | New Item & Participant | All |
|---|---|---|---|---|---|---|
| Majority Class / Chance | – | – | $50.0 \pm 0.0$ | $50.0 \pm 0.0$ | $50.0 \pm 0.0$ | $50.0 \pm 0.0$ |
| Reading Time | – | – | $59.0 \pm 0.4_{+++}$ | $58.9 \pm 1.0_{++}$ | $60.4 \pm 1.2$ | $59.0 \pm 0.5_{+++}$ |
| Log. Regression (Mézière et al., 2023) | Global | – | $62.4 \pm 0.3^{**}_{++}$ | $60.6 \pm 1.4_{++}$ | $60.8 \pm 1.6$ | $61.5 \pm 0.8^{*}_{+++}$ |
| BEyeLSTM - No Text | Fix. | – | $71.5 \pm 0.6^{***}_{+++}$ | $61.0 \pm 1.1$ | $61.5 \pm 1.5$ | $65.9 \pm 0.4^{***}$ |
| RoBERTa-Eye-W (Shubi et al., 2024) | Words | E+LF | $65.9 \pm 0.6^{***}_{+++}$ | $\mathbf{63.2 \pm 1.1}^{**}$ | $62.3 \pm 1.3$ | $64.4 \pm 0.6^{***}_{+++}$ |
| MAG-Eye (Shubi et al., 2024) | Words | E+LF | $52.1 \pm 0.3_{+++}$ | $52.3 \pm 0.4_{+++}$ | $51.5 \pm 0.4_{+++}$ | $52.1 \pm 0.2_{+++}$ |
| PLM-AS (Yang & Hollenstein, 2023) | Fix. Order | E | $58.6 \pm 0.4_{+++}$ | $59.5 \pm 0.5_{+++}$ | $57.5 \pm 0.9_{++}$ | $59.0 \pm 0.4_{+++}$ |
| Haller RNN (Haller et al., 2022) | Fix. | E | $61.3 \pm 0.8_{+++}$ | $61.5 \pm 0.8$ | $60.3 \pm 1.6$ | $61.3 \pm 0.4_{+++}$ |
| BEyeLSTM (Reich et al., 2022a) | Fix. | LF | $71.4 \pm 0.9^{***}_{+++}$ | $61.6 \pm 1.1$ | $62.2 \pm 1.3$ | $66.2 \pm 0.7^{**}_{+}$ |
| Eyettention (Deng et al., 2023) | Fix. | E+LF | $55.8 \pm 0.8_{+++}$ | $55.7 \pm 1.1_{+++}$ | $55.4 \pm 1.8_{+++}$ | $55.8 \pm 0.9_{+++}$ |
| RoBERTa-Eye-F (Shubi et al., 2024) | Fix. | E+LF | $\mathbf{74.7 \pm 1.2}^{***}$ | $63.1 \pm 1.3^{*}$ | $\mathbf{62.6 \pm 1.6}$ | $\mathbf{68.5 \pm 0.9}^{***}$ |
| PostFusion-Eye (Shubi et al., 2024) | Fix. | E+LF | $70.7 \pm 1.1^{***}_{+++}$ | $62.2 \pm 1.1^{*}$ | $61.4 \pm 1.0$ | $66.1 \pm 0.8^{***}_{+++}$ |

## 5.1 LOGISTIC ENSEMBLE

Ensemble models combine predictions from multiple individual models, and often yield better results than single models, especially when models have diverse predictive behaviors (Sagi & Rokach, 2018). To examine the appropriateness of this approach for our case, we compute the agreement between models' predictions, quantified with Cohen's Kappa (Cohen, 1960). Figure 3a depicts the pairwise agreement rates across models in the validation data, where we observe mostly moderate agreement rates. Figure 9 in Appendix E further suggests consistency across the three evaluation regimes. These outcomes indicate that while the examined models do not always differ substantially in accuracy, they likely capture different aspects of the eye movement data and the task.

Motivated by the results of this analysis, we introduce a simple **Logistic Ensemble**: a 10-feature logistic regression model that predicts the reading goal from the probability outputs of our 10 models. The model is trained for each fold on the validation set and tested on the corresponding three test sets. In Figure 3b we present the performance of the Logistic Ensemble, against the best performing model, RoBERTa-Eye-F, and the Reading Time baseline. We find that the Logistic Ensemble improves over the accuracy of RoBERTa-Eye-F in all the regimes, with an accuracy of 77.3 for New Items, 64.6 for New Participants, 64.3 for New Items & Participants, and 70.5 overall. The improvements over RoBERTa-Eye-F are statistically significant in all but the New Item & Participant regime, in which it is the only model that is statistically better than the Reading Time baseline. These performance improvements suggest that the information encoded by the different models is to some extent complementary.

## 6 WHAT MAKES THE TASK EASY OR HARD?

Having established that the prediction task at hand can be performed with a considerable degree of success, we now leverage the best performing model to obtain insights about the task itself. We do so by examining which trial features, which were not given to the model explicitly, affect the ability of the model to classify trials correctly. This analysis takes advantage of the rich structure and auxiliary annotations of the OneStop dataset.

We define 10 features that capture various aspects of the trial. These include the following reader features over the item:

1-3. **Reading time before, within, and after critical span**: These features are motivated by the findings of Malmaud et al. (2020) and Shubi & Berzak (2023) regarding faster reading

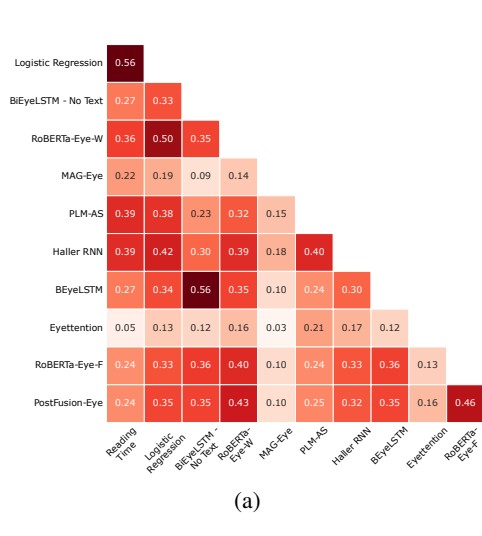 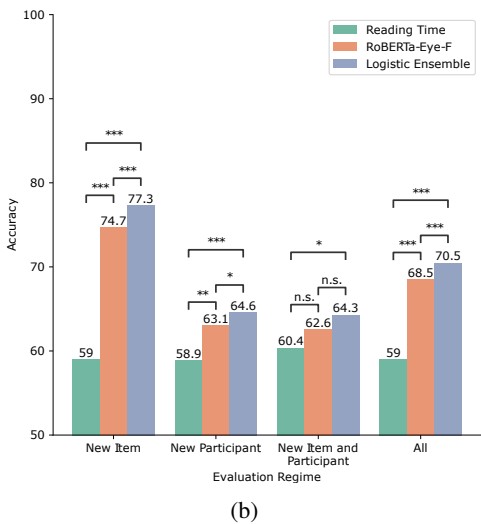

(a)                                                     (b)

Figure 3: (a) Pairwise Cohen's Kappa agreement between model predictions on the validation set. (b) Test accuracy of the Logistic Ensemble model compared to the best single model RoBERTa-Eye-F and the Reading Time baseline for the ordinary reading vs. information-seeking goal decoding task, using 10-fold cross-validation. Statistical significance is tested with a linear mixed effects model. In R notation: $is\_correct \sim model + (model \mid participant) + (paragraph \mid parag)$. '*' $p < 0.05$, '**' $p < 0.01$ and '***' $p < 0.001$.

    times in information seeking compared to ordinary reading, primarily before and after the critical span, as well as the reported classification results of the Reading Time baseline.

4. **Paragraph position** (1-54): Each participant reads 54 paragraphs, in a random article order. This feature captures the position of the paragraph in the experiment's presentation sequence. It is included as reading strategies can change as the experiment progresses (e.g. Meiri & Berzak (2024) show that readers become faster as the experiment progresses).

5. **Answered correctly**: this feature encodes whether after having read the passage, the participant answered the given reading comprehension question correctly. It captures participant-specific task difficulty and the extent to which the participant read the passage attentively.

We further include the following item (paragraph and question), reader-independent features:

6. **Paragraph length** (in words): this feature is chosen as we hypothesize that more data could lead to more accurate predictions for the item.

7. **Paragraph level** (Advanced / Elementary): is chosen as eye movements could be influenced by the difficulty level of the text, for example through differences in word frequency and surprisal (Singh et al., 2016; Hollenstein et al., 2022).

8. **Critical span start location** (relative position, normalized by paragraph length): Shubi & Berzak (2023) showed skimming-like reading patterns after processing task critical information in information seeking. We thus hypothesize that earlier appearance of task critical information could facilitate the ability to correctly identify information seeking reading.

9. **Critical span length** (normalized by paragraph length): we also hypothesize that less task critical information during information seeking could further aid distinguishing it from ordinary reading.

10. **Question difficulty** (percentage of participants who answered the question incorrectly): based on data from the Prolific QA experiment in Berzak et al. (2020). We include this feature as it can influence eye movements in information seeking, with harder questions potentially obscuring patterns of goal oriented reading.

In order to establish the relation of these features to task difficulty, we use a linear mixed effect model that uses these features to predict whether the trial prediction of the best performing model,

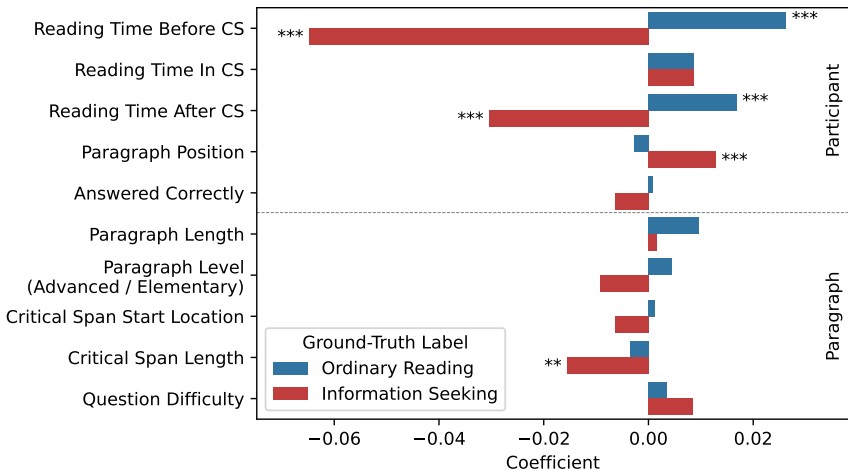

Figure 4: Coefficients from a mixed-effects model that predicts whether RoBERTa-Eye-F's prediction for a given trial is correct from properties of the trial. CS stands for the critical span, the portion of the paragraph that contains the information that is essential for answering the question correctly. Two models are fitted separately for ordinary reading and information seeking trials. Predictors are z-normalized. Depicted are the coefficients of the fitted models after a 10x Bonferroni correction, to mitigate the risk of false positives when testing multiple hypotheses simultaneously. '*' $p < 0.05$, '**' $p < 0.01$, '***' $p < 0.001$.

RoBERTa-Eye-F, was correct. In R notation:

$$\text{is\_correct} \sim 1 + \text{feat}_1 + \text{feat}_2 + \cdots + \text{feat}_{10} + (1 \mid \text{participant}) + (1 \mid \text{item}) + (1 \mid \text{evaluation regime})$$

where the random effects account for correlations in predictions within participants, items and evaluation regimes[2]. The advantage of a multiple regression model over separate univariate analyses of each feature, is the ability to tease apart the contribution of each feature above and beyond all other features. We fit this model separately on the information seeking and ordinary reading trials. To make the contributions of the features to prediction accuracy comparable, we normalize each feature to be a z-score (zero mean and unit variance). We then examine feature contribution via statistical significance, magnitude and sign of the corresponding coefficient. A significant coefficient for a feature indicates that it correlates with task difficulty, the absolute value determines its importance rank relative to other features, and the sign indicates the direction of the association.

The resulting feature coefficients are presented in Figure 4. In line with the findings of Shubi & Berzak (2023), we observe that the most prominent features for correctly classifying both information seeking and ordinary reading trials are reading time before and after the critical span. Slower readers in these regions are easier to correctly classify as ordinary reading and harder to correctly classify as information seeking. We also find, as hypothesized, that shorter critical spans facilitate correct classification of information seeking trials, presumably by making information seeking more targeted. Finally, paragraph position is significant in information seeking, suggesting that readers develop more efficient goal oriented reading strategies as they progress through the experiment. Overall, this analysis provides a highly interpretable characterization of task difficulty, with insights on reading behavior in both reading regimes.

## 7 RELATED WORK

The vast majority of the literature on the psychology of reading and its interfaces with computational modeling is concerned with ordinary reading. However, several studies did address goal-oriented (also referred to as task-based) reading. Most prior work focused on a small number of canonical tasks: skimming, speed reading and proofreading. Several studies found different eye movement patterns in these tasks as compared to ordinary reading (Just et al., 1982; Kaakinen & Hyönä, 2010;

---

[2]Random effects structure is simplified not to include slopes due to model convergence issues.

Schotter et al., 2014; Strukelj & Niehorster, 2018; Chen et al., 2023). Rayner & Raney (1996) examined differences between ordinary reading and searching through the text for a target word. Prior work also analyzed eye movements during human linguistic annotation, often used for generating training data for NLP tools, such as annotation of named entities (Tomanek et al., 2010; Tokunaga et al., 2017). Differences in reading patterns were also found when readers were asked to take different perspectives on a given text (Kaakinen et al., 2002) or given different sets of learning goals (Rothkopf & Billington, 1979). We further note that information seeking has also been previously examined in visual search (Jang et al., 2014a;b; Sharma et al., 2023).

Our work is closest to Malmaud et al. (2020), Hahn & Keller (2023) and Shubi & Berzak (2023) who analyzed eye movement differences between ordinary reading and information seeking, where the information seeking goal is formulated using a reading comprehension question. All three studies found substantial differences in fixation and saccade patterns in information seeking as compared to ordinary reading, in particular before, within and after the text portions that are critical for the information seeking task. Here, we build on these findings, and examine whether these differences can be leveraged for automatic discrimination between these two reading regimes.

While the above studies focus primarily on descriptive data analysis, Hollenstein et al. (2023) took a predictive approach and attempted to automatically classify the reading task from eye movement features. In this study, participants read single sentences from the ZuCo corpus (Hollenstein et al., 2020), and engaged either in ordinary reading or in an annotation task where they had to determine the presence of one of seven semantic relations in the sentence. While this benchmark is conceptually related to the current work, it is restricted in scope and ecological validity. First, the benchmark contains only 18 participants, which makes generalization and reliable evaluations highly challenging. More fundamentally, the framework is limited by the nature of the tasks, which focus on highly specialized linguistic annotations that are not performed by readers in everyday life. If viewed through the lens of question answering, the setup comprises only seven possible questions that concern the presence of a semantic relation in the sentence. In the current study we take a different and more general stance on task based reading, with unrestricted questions that are more representative of the tasks commonly pursued by readers. Finally, additional related lines of work used eye movements in reading for prediction of other types of reader cognitive state, such as reading comprehension (Reich et al., 2022b; Shubi et al., 2024), as well as inferring properties of the text such document type (Kunze et al., 2013) and readability level (González-Garduño & Søgaard, 2017).

## 8 SUMMARY AND DISCUSSION

Is it possible to decode reader goals from eye movements? We address this question by examining the possibility to automatically discriminate between ordinary reading and information seeking at the challenging granularity level of a single paragraph. We find that it is indeed possible to perform this task with considerable success, and show that eye movements provide valuable information for this task, above and beyond reading time. We further find that different models capture different aspects of the signal and the task, and can be successfully combined with ensembling, which yields the strongest results for our task. Our error analysis leverages the models to further reveal new insights on the factors that determine task difficulty. Our evaluation setup which addresses different aspects of model generalization provides an infrastructure for future work on the proposed task.

Our study has a number of important limitations. The information seeking tasks are over individual paragraphs that span 3-10 lines of text. This leaves out shorter texts (e.g. single sentences) as well as longer texts. It is also restricted to newswire texts, and does not include texts from other genres. Finally, new datasets for the information seeking task, other types of tasks, additional populations (e.g. second language readers, younger and older participants), and datasets in languages other than English are all needed in order to study goal decoding more broadly.

While the current work takes a first step in addressing the proposed task, ample room for performance improvements remains for future work, especially in the new participant regimes. New strategies for modeling eye movements with text are likely needed to fully exploit the potential of eye movements for this task. Furthermore, the addressed task is fundamentally limited in that it does not distinguish between different information seeking tasks. A natural direction for future work could address decoding of the specific question that was presented to the participant in the information seeking regime.

## 9 ETHICS STATEMENT

This work uses eye movement data collected from human participants. The data was collected by Malmaud et al. (2020) under an institutional IRB protocol. All the participants provided written consent prior to participating in the eyetracking study. The data is anonymized.

It has previously been shown that eye movements can be used for user identification (e.g. Bednarik et al., 2005; Jäger et al., 2020). We do not perform user identification in this study, and emphasize the importance of not storing information that could enable participant identification in future applications of goal decoding. We further stress that future systems that automatically infer reader goals are to be used only with explicit consent from potential users to have their eye movements collected and analyzed for this purpose.

## 10 REPRODUCIBILITY STATEMENT

We describe the model training and selection procedure, the evaluation protocol, and the hardware and software specifications in Section 4.2. The hyperparameter search space for each model is described in Appendix C. The dataset used in the experiments is described in Section 4.1 and appendix B. The source code, which implements the models, experimental procedures, and analysis, is available at: `https://anonymous.4open.science/r/Decoding-Reading-Goals-from-Eye-Movements/`. Data will be made publicly available.

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

# APPENDIX

## A MODEL DIAGRAMS

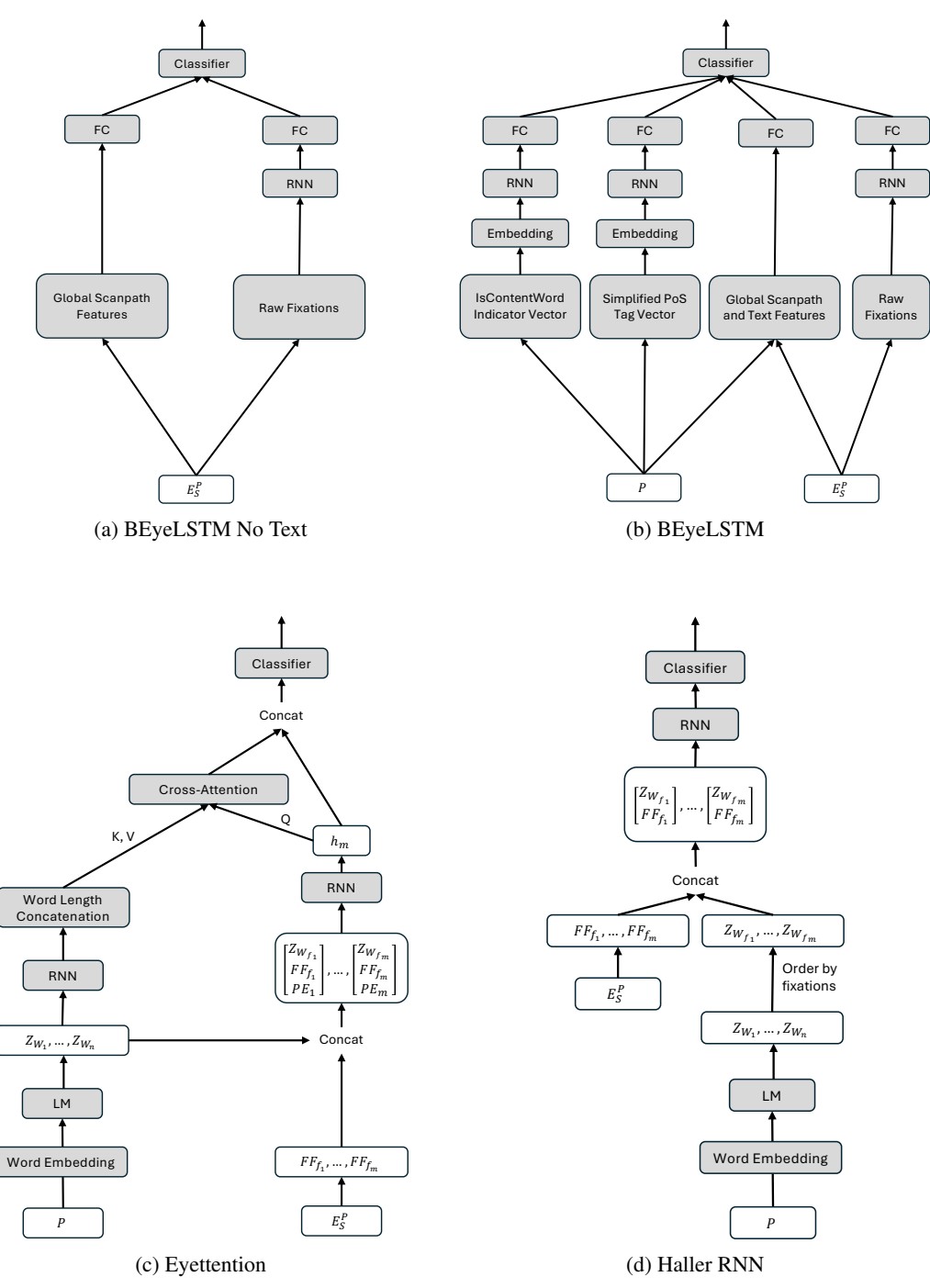

Figure 5: Visualization of the different model architectures (Part 1). $P$ represents the paragraph, $E_S^P$ the eye movements of participant $S$ on $P$. $LM$ stands for a language model, and $FC$ for fully connected layers. $FF_{f_i}$ stands for the fixation features and $w_{f_i}$ for the word corresponding to the $i$-th fixation respectively.

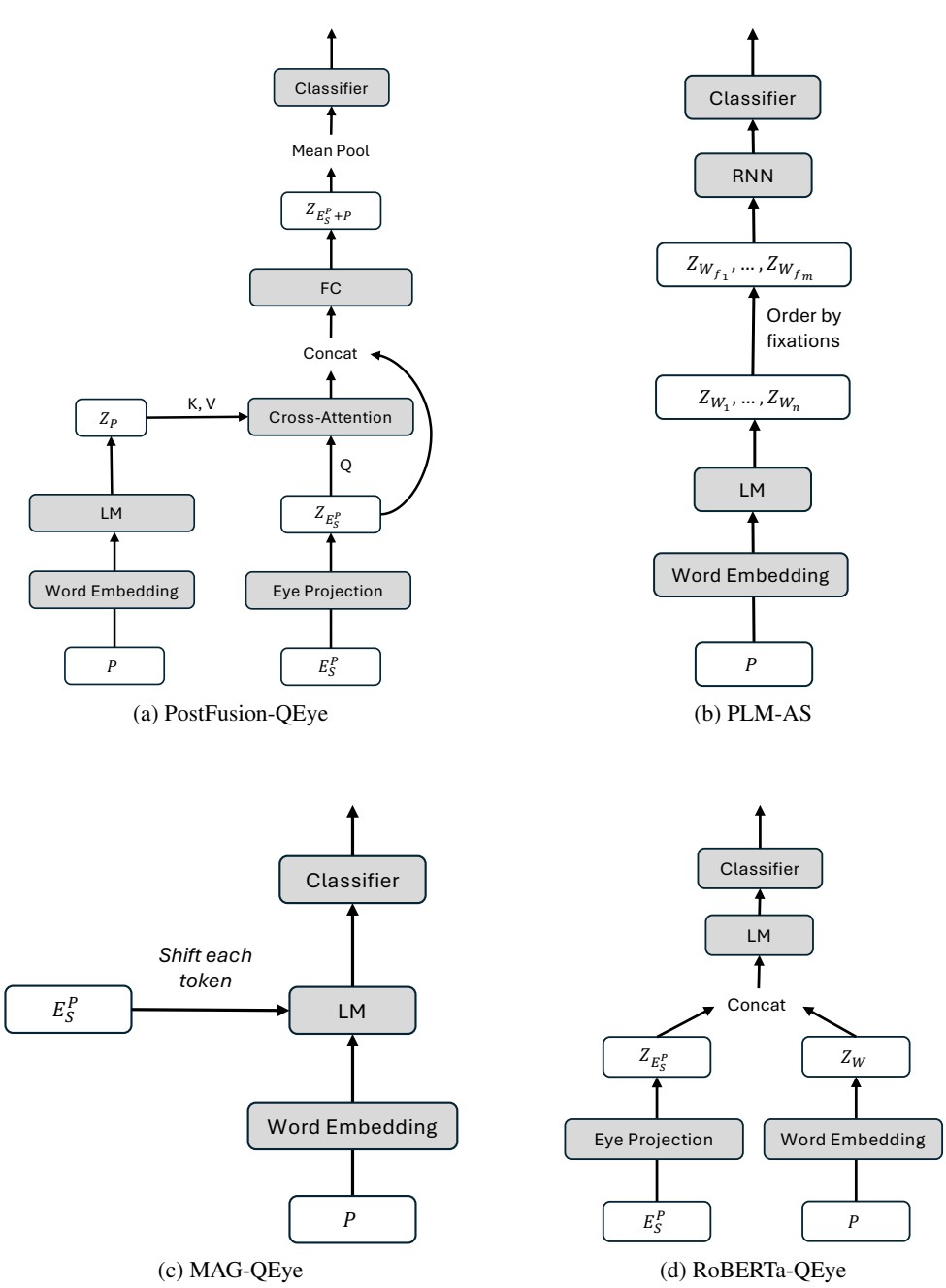

Figure 6: Visualization of the different model architectures (Part 2). $P$ represents the paragraph, $E_S^P$ the eye movements of participant $S$ on $P$. $LM$ stands for a language model, and $FC$ for fully connected layers. $FF_{f_i}$ stands for the fixation features and $w_{f_i}$ for the word corresponding to the $i$-th fixation respectively.

# B ONESTOP EYE MOVEMENTS DATASET - ADDITIONAL DETAILS

The textual data of OneStop consists of 162 paragraphs, 486 questions, and 972 unique paragraph–level–question triplets. The mean length of Elementary paragraphs is 97 words (37 before the critical span, 30 inside it, and 30 after it), and of Advanced paragraphs 120 words (48 before the critical span, 34 inside it, and 38 after it). Each question has 20 responses, 10 for the Advanced version and 10 for the Elementary version. The mean experiment duration is approximately one hour. The raw millisecond gaze location is pre-processed into fixations and saccades using the SR Data Viewer software (`v4.3.210`).

Table 2: An example of a OneStopQA paragraph (Advanced and Elementary version) along with one of its three questions. The critical span is marked in bold red. Adapted from Berzak et al. (2020).

| | |
|---|---|
| **Advanced** | A major international disagreement with wide-ranging implications for global drugs policy has erupted over the right of Bolivia's indigenous Indian tribes to chew coca leaves, the principal ingredient in cocaine. **Bolivia has obtained a special exemption from the 1961 Single Convention on Narcotic Drugs, the framework that governs international drugs policy, allowing its indigenous people to chew the leaves.** Bolivia had argued that the convention was in opposition to its new constitution, adopted in 2009, which obliges it to "protect native and ancestral coca as cultural patrimony" and maintains that coca "in its natural state ... is not a dangerous narcotic." |
| **Elementary** | A big international disagreement has started over the right of Bolivia's indigenous Indian tribes to chew coca leaves, the main ingredient in cocaine. This could have a significant effect on global drugs policy. **Bolivia has received a special exemption from the 1961 Convention on Drugs, the agreement that controls international drugs policy. The exemption allows Bolivia's indigenous people to chew the leaves.** Bolivia said that the convention was against its new constitution, adopted in 2009, which says it must "protect native and ancestral coca" as part of its cultural heritage and says that coca "in its natural state ... is not a dangerous drug." |
| **Question Answers** | **What was the purpose of the 1961 Convention on Drugs?**
A Regulating international policy on drugs
B Discussing whether indigenous people in Bolivia should be allowed to chew coca leaves
C Discussing the legal status of Bolivia's constitution
D Negotiating extradition agreements for drug traffickers |

Ordinary Reading          Information Seeking

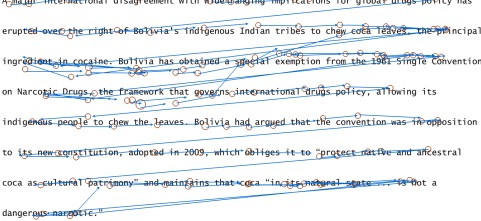
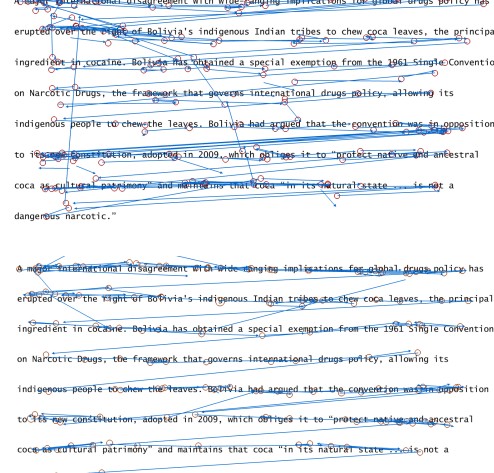

Figure 7: Examples of eye movements over a single passage; left: ordinary reading, right: information seeking. Circles represent fixations, and lines represent saccades.

## C    MODEL TRAINING AND HYPERPARAMETERS

Since the models we use were developed for different tasks and datasets, we conducted a hyperparameter search for each model. The search space for each model is described below. In all cases, it includes the optimal parameters reported in the work that introduced the model, extended to provide a fair comparison between models.

For all neural models we train with learning rates of $\{0.00001, 0.00003, 0.0001\}$ and dropout of $\{0.1, 0.3, 0.5\}$ following Shubi et al. (2024). Additionally, for all models that make use of word embeddings, we include both frozen and unfrozen language model variants in the search space.

- For **Logistic Regression**, the search space for the regularization parameter C is $\{0.1, 5, 10, 50, 100\}$, with and without an L2 penalty.
- Following (Reich et al., 2022a), for **BEyeLSTM** and **BEyeLSTM-No Text**, the search space consists of learning rates $\{0.001, 0.003, 0.01\}$, embedding dimensions $\{4, 8\}$ and hidden dimensions $\{64, 128\}$.
- For **MAG-Eye** the search space for injection layer index is: $\{0, 11, 23\}$.
- Following Yang & Hollenstein (2023), we train **PLM-AS** and **Haller RNN** with dropout rate search space of 0.1, and for PLM-AS, we use LSTM layer counts of $1, 2$. Additionally, as in (Haller et al., 2022), we search over LSTM hidden layer sizes of $10, 40, 70$. For PLM-AS, the LSTM hidden layer size is fixed at 1024 to match the LM's dimensionality in Yang & Hollenstein (2023).
- For **Eyettention**, we also train with a learning rate of 0.001 and dropout of 0.2, as done in (Deng et al., 2023)
- For **PostFusion-Eye**, the 1D convolution layers have a kernel size of three, stride 1 and padding 1.

We train the deep-learning based models for a maximum of 40 epochs, with early stopping after 8 epochs if no improvement in the validation error is observed. Following Liu et al. (2019); Mosbach et al. (2021); Shubi et al. (2024) we use the AdamW optimizer Loshchilov & Hutter (2018) with a batch size of 16. MAG-Eye, RoBERTa-Eye and PostFusion-Eye use a linear warm-up ratio of 0.06, and a weight decay of 0.1. We standardize each eye movements feature using statistics computed on the training set, to zero mean unit variance. Models are implemented based on code from (Shubi et al., 2024).

# D  ADDITIONAL RESULTS

Below we present the test set ROC curves across the ten cross-validation splits and their corresponding AUROC scores (mean and standard deviation). We also provide accuracy results for the validation set.

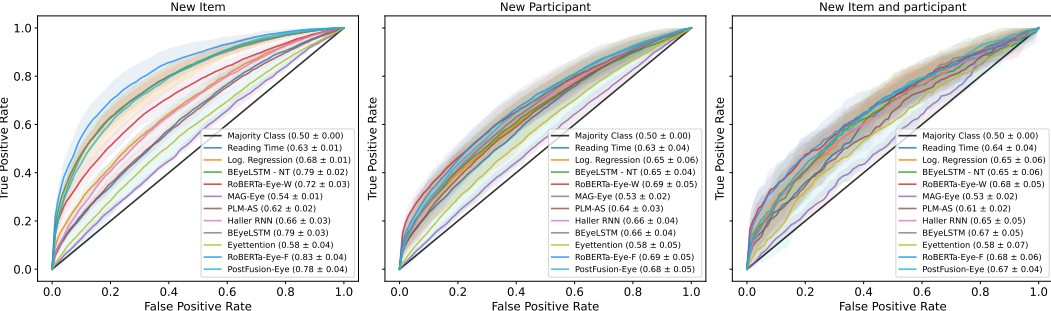

Figure 8: ROC Curves by model and evaluation regime. Each curve represents a different model across the ten cross-validation splits, with the corresponding AUROC scores (mean and standard deviation) provided in the legend.

Table 3: Validation accuracy results aggregated across 10 cross-validation splits. 'Emb' stands for word embeddings, 'Ling. Feat.' for linguistic word properties. Model performance is compared to the Reading Time baseline using a linear mixed-effects model. In R notation: $is\_correct \sim model + (model \mid participant) + (model \mid paragraph)$. Significant gains over this baseline are marked with '*' $p < 0.05$, '**' $p < 0.01$ and '***' $p < 0.001$ in superscript, and significant drops compared to the best model in each regime are marked in subscript with '+'.

| Model | Gaze Representation | Text Representation | New Item | New Participant | New Item & Participant | All |
|---|---|---|---|---|---|---|
| Majority Class / Chance | None | None | $50.0 \pm 0.0$ | $50.0 \pm 0.0$ | $50.0 \pm 0.0$ | $50.0 \pm 0.0$ |
| Reading Time | None | None | $58.9 \pm 0.5_{+++}$ | $58.9 \pm 1.0_{+++}$ | $60.4 \pm 1.3_{++}$ | $58.9 \pm 0.5_{+++}$ |
| Logistic Regression | Global | None | $62.6 \pm 0.3^{**}_{++}$ | $60.6 \pm 1.5_{+++}$ | $61.0 \pm 1.8_{++}$ | $61.6 \pm 0.8^{*}_{+++}$ |
| BiEyeLSTM - No Text | Fixations | None | $73.2 \pm 0.6^{***}_{+}$ | $64.9 \pm 1.0$ | $65.1 \pm 1.4$ | $68.8 \pm 0.5^{***}$ |
| RoBERTa-Eye-W | Words | Emb + Ling. Feat. | $66.3 \pm 0.4^{***}_{+}$ | $\mathbf{65.4 \pm 1.1}^{**}$ | $64.1 \pm 1.0$ | $65.8 \pm 0.5^{***}_{+}$ |
| MAG-Eye | Words | Emb + Ling. Feat. | $53.7 \pm 0.2_{+++}$ | $53.5 \pm 0.5_{+++}$ | $52.6 \pm 0.6_{+++}$ | $53.5 \pm 0.2_{+++}$ |
| PLM-AS | Fixations Order | Emb | $59.1 \pm 0.5_{+++}$ | $61.1 \pm 0.7_{+++}$ | $58.6 \pm 0.9_{+++}$ | $60.1 \pm 0.4_{+++}$ |
| Haller RNN | Fixations | Emb | $62.5 \pm 0.4_{+++}$ | $62.4 \pm 1.1_{++}$ | $61.4 \pm 1.4_{++}$ | $62.3 \pm 0.5_{+++}$ |
| BEyeLSTM | Fixations | Ling. Feat. | $72.3 \pm 0.6^{***}_{+}$ | $65.0 \pm 1.3$ | $\mathbf{66.1 \pm 1.2}$ | $68.5 \pm 0.6^{***}$ |
| Eyettention | Fixations | Emb + Ling. Feat. | $56.4 \pm 0.8_{+++}$ | $56.6 \pm 0.9_{+++}$ | $58.6 \pm 1.1_{+++}$ | $56.6 \pm 0.5_{+++}$ |
| RoBERTa-Eye-F | Fixations | Emb + Ling. Feat. | $\mathbf{76.5 \pm 1.1}^{***}$ | $64.8 \pm 1.3^{*}$ | $65.9 \pm 1.4$ | $\mathbf{70.3 \pm 0.7}^{***}$ |
| PostFusion-Eye | Fixations | Emb + Ling. Feat. | $72.8 \pm 0.8^{***}_{+}$ | $64.6 \pm 1.4^{*}$ | $63.6 \pm 1.8$ | $68.4 \pm 0.9^{***}_{++}$ |

# E PAIRWISE AGREEMENT BETWEEN MODELS BY EVALUATION REGIME

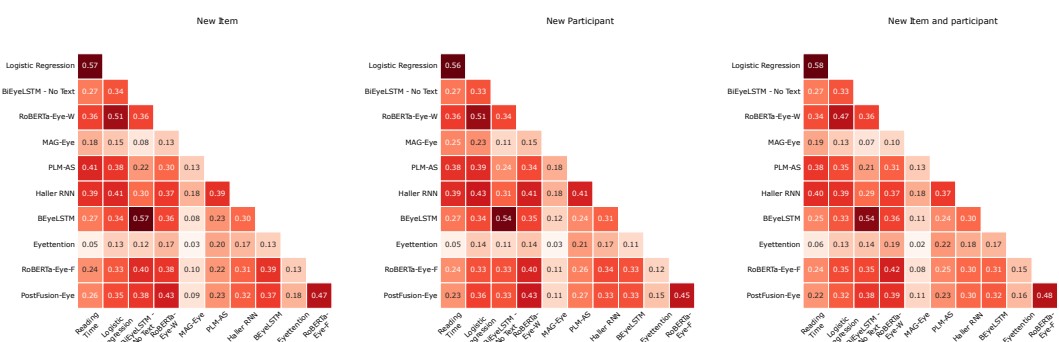

Figure 9: Pairwise Cohen's Kappa agreement between model predictions on the validation set by evaluation regime.

