# OpenReview forum: "Decoding Reading Goals from Eye Movements"
_ICLR.cc/2025/Conference — Submitted to ICLR 2025_

### Official Review · Reviewer_Y9QK · 2024-10-29

**Soundness:** 3
**Presentation:** 3
**Contribution:** 2
**Rating:** 6
**Confidence:** 4

**Summary:**

The authors investigate automatic classification of reading behaviour into information seeking and "ordinary reading". They recorded a new eye tracking dataset for this task and evaluated several relevant approaches.

**Strengths:**

The article is very well written and easy to follow.
The experiments are extensive, sound, and deliver some meaningful insights.
The dataset can be a valuable resource for the community.

**Weaknesses:**

The novelty is limited: the author conduct a number of evaluations of existing methods an a new dataset.
The novelty of the task is also very marginal and incorrectly portrayed as the authors miss highly relevant previous work. Xiuge et al. (2023) distinguished deep from skim reading using eye tracking, which is very similar to what the authors claim as novelty.

Further relevant previous work that should be discussed is by Kunze et al. (2013) who distinguished different document types.
It would also be meaningful to compare the reading goal recognition task to similar tasks that were investigated with eye tracking data. One example is informational versus navigational intent classification (Sharma et al., 2023).

In my opinion, if the authors re-formulate their contribution, taking all the relevant related work into account, the paper could still be interesting. I would not see it at ICLR due to the limited contribution w.r.t. to deep learning, but rather at a more HCI-focused venue such as ETRA, ICMI, IUI,...

References:
-----------

Chen, Xiuge, et al. "Characteristics of Deep and Skim Reading on Smartphones vs. Desktop: A Comparative Study." Proceedings of the 2023 CHI Conference on Human Factors in Computing Systems. 2023.

Kunze, Kai, et al. "I know what you are reading: recognition of document types using mobile eye tracking." Proceedings of the 2013 international symposium on wearable computers. 2013.

Sharma, Mansi, et al. "Implicit Search Intent Recognition using EEG and Eye Tracking: Novel Dataset and Cross-User Prediction." Proceedings of the 25th International Conference on Multimodal Interaction. 2023.

**Questions:**

no questions

---

> ### Author Response · Authors · 2024-11-19
> **Reply to Reviewer Y9QK**
>
> Thank you very much for your review and the positive comments regarding the paper and the experiments.
>
> We have added all three proposed references, which have resulted in better positioning of our paper relative to prior work.
>
> We would like to elaborate on the novelty of our work with respect to these papers:
>
> **Task**: the task of **automatically distinguishing between ordinary and information seeking reading has not been addressed in previous work**. In particular, the references you provide are indeed related to the current work but deal with different tasks:
>
> - Chen et al. (2023): compares features that were previously found to discriminate between skimming and deep reading in smartphone vs desktop monitors. We discuss a number of studies that deal with skimming in the paper. Leaving aside the specific question of the paper (smartphone vs desktop) which is tangential to our study, skimming vs deep reading is related to our study in that these are two different reading regimes that can be distinguished based on eye movements. Shubi and Berzak (2023) have also shown that information seeking involves eye movement behavior that resembles skimming after participants have found and processed question critical information. However, skimming and information seeking are different types of tasks. One way to think about the difference is that in skimming vs deep reading, the task is purely *procedural.* It focuses on a general manner of reading (skimming or not) and does not strongly depend on the text (one can skim any given text with similar eye movement behavior e.g. shorter fixations, more skips, longer saccades etc.). In our case, the task is *semantic*, the information seeking is specific to the text. Differently from skimming, it is manifested differently for different questions (and of course texts and readers), with more complex eye movements patterns (information search, critical information processing, post critical span wrap up) making our task fundamentally different and more challenging than identifying skimming. A different way to think about it is the following: skimming is a special and simple case of information seeking, where the reader has **only one possible goal** that can be formulated using the question ”Please skim the text” (or similar). Our study addresses hundreds of text specific tasks / questions and provides a conceptual and practical open ended setup were the reader can have **arbitrary goals** with respect to the text.
> - Kunze et al. (2013): is a pilot study with 8 participants on distinguishing between two document types (left to right top to bottom reading vs top to bottom right to left reading). While related in the sense that there is a prediction task from eye movements in reading, here too the task is fundamentally different: predicting **a property of the text**, while we predict a **property of the cognitive state of the reader**.
> - Sharma et al. (2023): distinguishes glancing into a scene (navigational intent) or searching for a target object (informational intent), which is indeed conceptually related to our study. Importantly however,  this study was conducted over images where the task is **visual search,** which **does not involve** **reading**.
>
> **Paper contribution**: we would like to point out that this paper is in the applications track, and as such focuses more on defining, benchmarking, and providing an **analysis and insights for the task at hand**. Please note that we do provide new insights about existing models for eye movements and text which also lead to **a new model ensemble**. An additional contribution of this paper is in **model interpretability**: we show how models can be leveraged for characterizing critical features for the task at hand via mixed effects modeling, a technique that to our knowledge has not been previously used for this purpose and can be applied to analysis of other classification tasks.
>
> To summarize, while we acknowledge that related work that involves eyetracking clearly exist, and are very thankful for the provided references which we added to the paper, we believe that they do not take away from the novelty of the task addressed in our paper. The paper further offers several substantial contributions in model analysis and interpretability leading to better understanding of the task.
>
> Given these contributions, and your otherwise positive view of the paper, we kindly ask to consider raising the score of the paper.

---

> > ### Comment · Reviewer_Y9QK · 2024-11-22
> > **Response**
> >
> > Thank you for your response!
> > I am raising my score.
> >
> > The positioning with respect to previous work has indeed improved thanks to the response. I still think the novelty is moderate only - it would definitely be helpful to show empirically how the task addressed by the authors is different or more challenging compared to skimming vs. deep reading classification. However their theoretical arguments certainly are meaningful.

---

> > > ### Author Response · Authors · 2024-11-25
> > >
> > > Thank you very much!

---

### Official Review · Reviewer_v1ny · 2024-11-04

**Soundness:** 2
**Presentation:** 3
**Contribution:** 1
**Rating:** 3
**Confidence:** 5

**Summary:**

This paper adopts an existing dataset and existing methods to test reading goal prediction, which consists in predicting if a human is performing ordinary reading or information seeking (binary classification task). In addition to testing existing methods, the authors introduce a model ensemble which shows improved performance, highlighting that different models capture different information. Finally, the authors study how different trial features affect the model's prediction. Reading time before/after critical span, critical span length and paragraph position seem to be the most crucial features.

**Strengths:**

Originality. The paper proposes an interesting research question, i.e. if one can distinguish between two reading tasks based on eye movements (+ text). This can be seen as an extension of previous work, in particular [1]. The authors suggest that their paper has broader scope and ecological validity than [1]. While this represents an important and valid extension, it is not particularly stark in originality. The methodologies are all adapted from previous works, as well as the data. While taking a different perspective, the conceptual framework is almost identical to previous work [1,2].

Quality. The methods used are appropriate and the analysis performed is appropriate, well presented, and interpreted. The authors released their code, which together with the implementation details provided in the paper should make the results reproducible.

Clarity. The paper is well written and exhaustively reports implementation details. To further improve clarity, I would add a more clear explanation of the difference between the proposed task (reading goal classification) and reading comprehension.

Significance. The paper represents an interesting contribution to the field of psycholinguistics, exploring a new task. The analysis in Section 6 highlights important features to consider when studying eye movements on text.

[1] Hollenstein, Nora, et al. "ZuCo 2.0: A Dataset of Physiological Recordings During Natural Reading and Annotation." Proceedings of the Twelfth Language Resources and Evaluation Conference. 2020.
[2] Shubi, Omer, et al. "Fine-Grained Prediction of Reading Comprehension from Eye Movements." arXiv preprint arXiv:2410.04484 (2024).

**Weaknesses:**

- As discussed above, the paper is an interesting extension of previous work but does not particularly shine in terms of originality.

- The authors do not discuss limitations of current methods or dataset. The authors briefly mention the need for different tasks and datasets in Section 8, but I believe a more structured and systematic discussion of limitations is necessary. For example:
  - The paragraphs considered in the study are all short. Do the findings generalise to longer text?
  - Being taken from newspaper articles, all the paragraphs are expository - they provide information. Do the authors expect to find similar results for, e.g., narrative texts written to entertain?
  - There are differences between different reading comprehension tasks [1]. For example, students that read a text when doing an exam have likely a different behaviour than students that read just to understand the text. Are these differences relevant to this paper?

- The scientific implications of the authors’ findings can be better discussed. Based on their findings, how can we improve current methods? How can we fully exploit the potential of eye movements for such a task?

- While the authors state that reading is an ubiquitous and essential skill, the paper lacks an explanation of why reading goal prediction is an important task to study and which are the applications or areas that will benefit the most from that.

- Overall, this paper presents interesting research in psycholinguistics. However, I believe its contributions are not strong and original enough to be published at ICLR. In addition, the topic would probably fit better a psycholinguistics or computational linguistics venue.

[1] Mar, Raymond A., et al. "Memory and comprehension of narrative versus expository texts: A meta-analysis." Psychonomic Bulletin & Review 28 (2021): 732-749.

**Questions:**

see above

---

> ### Author Response · Authors · 2024-11-19
> **Reply to Reviewer v1ny - Part 1**
>
> Thank you very much for the detailed review and the positive comments.
>
> Please find responses to your comments and subsequent edits to the paper below.
>
> > As discussed above, the paper is an interesting extension of previous work but does not particularly shine in terms of originality.
>
> We would like to clarify why our work is not merely an extension of Hollenstein et al. 2023. This paper distinguishes ordinary reading from **specialized linguistic annotation of semantic relations**. First, we discuss the differences from this paper at length in the related work section. To put things simply, while this work is indeed related to the current study, it **does not address information seeking** in its standard everyday interpretation. Instead, it asks participants to perform a specific linguistic annotation of semantic relations. While both papers deal with what can be superficially named as “tasks” these are fundamentally different types of tasks. As we further note in the paper, our setup is much broader and much more relevant for everyday reading situations (people regularly search for information in text, while they almost never perform specialized linguistic annotations).
>
> - Relation between our task and reading comprehension
>
> It is indeed a highly interesting question. In the paper we empirically examine the relation between our task and reading comprehension in the feature analysis through the features “answered correctly” (did the participant answer the reading comprehension correctly) and “question difficulty” (the overall difficulty of the question measured across participants). As it turns out (Fig 4), these factors do not significantly influence the difficulty of distinguishing information seeking from ordinary reading.
>
> > The authors do not discuss limitations of current methods or dataset. The authors briefly mention the need for different tasks and datasets in Section 8, but I believe a more structured and systematic discussion of limitations is necessary.
>
> Thank you for this important suggestion and for the reference. We have extended the discussion to address your comments. Among others, we now discuss the text length and text genre limitations of the dataset. Regarding the length of the paragraphs, we now further report the minimum, maximum and standard deviation from the mean to provide a better sense of the variability in the length (from 3 - 10 lines of text). Given the current analysis that shows robustness to the paragraph length and the variety of topics in the texts, we expect the results to generalize to longer texts and to other genres. Showing this empirically would require collecting new eyetracking data which we intend to do in the future. Regarding your comment on the setup (classroom exam vs at home) this could indeed further influence reading patterns and would be an interesting question (and highly challenging question to study empirically!) for future eyetracking studies.
>
> > The scientific implications of the authors’ findings can be better discussed. Based on their findings, how can we improve current methods? How can we fully exploit the potential of eye movements for such a task?
>
> We have edited the discussion on scientific implications. Among others, these stem from the comparison of the models and are manifested in the finding that they capture different aspects of the task (as a natural conclusion we show that an ensemble model indeed improves the performance compared to the best model). We further provide an evaluation infrastructure (new items, new subjects, both) that will support rigorous evaluations of the generalization ability of future models. Each such regime corresponds to a real world scenario in terms of which prior eyetracking data is available to the model. Our feature analysis highlights important and fine-grained aspects of items and participants that future models need to capture effectively in order to improve classification accuracy.

---

> > ### Author Response · Authors · 2024-11-19
> > **Reply to Reviwer v1ny - Part 2**
> >
> > > While the authors state that reading is an ubiquitous and essential skill, the paper lacks an explanation of why reading goal prediction is an important task to study and which are the applications or areas that will benefit the most from that.
> >
> > This is an important point. While our current focus is on exploring a basic scientific question, we do see potential applications for our work. For example, in various educational scenarios it can be beneficial to monitor in real time how effectively students are engaging with a given reading comprehension task (e.g. from the confidence of the classifier that the given eye movements are information seeking). This in turn, can lead to targeted interventions and assistance with developing more effective reading and information seeking skills. It can also find uses in user facing applications where if the system detects an information seeking goal (e.g. finding out the price, or some other property of a product) the content can be adjusted to highlight information that is relevant to the user’s needs. Finally, an ability to discriminate goal seeking from ordinary reading can be used for assistive technologies for special populations. For example if we can detect that an elderly user is searching for information on a municipal or governmental website in real time, one can provide special assistance in such cases.

---

> > ### Comment · Reviewer_v1ny · 2024-12-02
> >
> > While I appreciate the changes made and the explanations given, I still do not think that the technical contribution above the bar for a top ML conference as ICLR. The paper makes no technical/method contribution, which makes it a good fit for one of the various HCI venues but - in my opinion, unsuited for ICLR.

---

> > > ### Author Response · Authors · 2024-12-03
> > >
> > > Thank you for your engagement. Respectfully, we disagree. As we outline in the summary above (and of course in the paper), beyond the task itself, the paper has several technical contributions, including a new (general purpose) mixed-effects method for model and task interpretability, a fine grained error analysis, adaptations and analysis of several state-of-the-art multimodal models for eye movements that were originally designed of other tasks, and a new and effective model ensemble. All are valid contributions for a top ML venue. We further stress that this paper is in the applications for cognitive science track, and as such also focuses on leveraging ML methods for scientific questions in cognitive science and psycholinguistics.

---

> ### Author Response · Authors · 2024-11-25
>
> Dear reviewer v1ny,
>
> Thank you again for your detailed and constructive feedback.
>
> We would like to draw your attention to our reply and to the subsequent manuscript revisions which address and integrate your comments.
>
> Given these revisions and your otherwise positive assessment of the paper, we kindly ask to consider raising your score for the paper.

---

### Official Review · Reviewer_RdFV · 2024-11-05

**Soundness:** 3
**Presentation:** 3
**Contribution:** 3
**Rating:** 6
**Confidence:** 4

**Summary:**

The paper explores a novel task of predicting reading goals - specifically distinguishing between information-seeking and ordinary reading - based on eye movement data. Utilizing an extensive dataset and advanced machine learning models, the authors demonstrate that eye movement patterns contain valuable signals for goal decoding. The study systematically evaluates several state-of-the-art models, including both eye-movement-only and multimodal (eye movements + text) approaches, and introduces an ensemble model that further improves prediction accuracy. The paper provides an in-depth error analysis, identifying key challenges in goal decoding and highlighting textual and participant-specific factors affecting classification difficulty.

**Strengths:**

Originality: The study introduces a novel problem - decoding reading goals from eye movements - that has not been widely explored. This new application area could encourage further research in cognitive science and multimodal data analysis.
Quality: The use of diverse models and a comprehensive evaluation protocol ensures the quality and reliability of the findings. The error analysis is particularly valuable, as it provides insights into the factors that influence classification success, such as the presence of critical spans and reading time.
Clarity: The paper explains the decoding task well and provides a logical flow of ideas. The inclusion of various model types and a mixed-effects model analysis demonstrates a comprehensive and thought-out approach.
Significance: This study lays a foundation for understanding goal-oriented reading behaviors. While applications are currently exploratory, the findings could aid cognitive science researchers and developers of educational tools who aim to personalize learning experiences based on user behavior.

**Weaknesses:**

Complexity in Model Descriptions: Some model descriptions lack clarity, especially in multimodal integration approaches. Providing visual diagrams or more intuitive breakdowns could make these sections more understandable.
Limited Scope of Goal Types: The study only explores two reading goals (information seeking and ordinary reading). Extending this research to other goals (e.g., skimming, proofreading) could make the findings more broadly applicable.

**Questions:**

Could the authors clarify how reading goal classification would perform in real-world applications where eye-tracking calibration is variable, such as web-based eye tracking?
Have the authors considered testing whether other task-specific goals (e.g., skimming) could also be accurately predicted using these models?

---

> ### Author Response · Authors · 2024-11-19
> **Reply to Reviewer RdFV**
>
> Thank you for the thorough and positive review!
>
> We start by replying to the questions and then to the other comments.
>
> * Performance in real-world applications
>
> > Could the authors clarify how reading goal classification would perform in real-world applications where eye-tracking calibration is variable, such as web-based eye tracking?
> >
>
> First and foremost, throughout the paper we make the distinction between the three test regimes which we believe is crucial for real-world applications. The new participants regime corresponds to a situation in which we have previous responses on the given passage, but not from the participant. This corresponds to a standard examination setup where we are interested in predicting behavior on a given set of materials for participants that haven't read them yet. In the new items regime we have eye movement recordings from participants for some prior passages, and we evaluate on previously unseen passages. This corresponds to e.g. an e-learning scenario where eye movement data is collected continuously from the participants while new items are presented. The new item and new participant regime is arguably the most interesting and flexible, as it allows making predictions for an arbitrary reader reading an arbitrary passage in a ‘zero-shot’ manner. As reflected in the results, this is indeed the most challenging regime, and only our ensemble approach achieves statistically significant improvement over a strong reading speed baseline.
>
> Second, in the future it would definitely be interesting to gather web-cam based eye tracking data for real world scenarios in which only lower quality eyetracking is available. With current low grade eyetracking technology (such as laptop and phone webcams) precise alignment between fixations and words is difficult to achieve, so, for current real-world applications, models that do not require this alignment are preferable. Our experiments show that ‘BEyeLSTM - No Text’ achieves relatively strong results, and we therefore think that currently eye movements-only models are the go-to solution in such cases.
>
> * Other task-specific goals
>
> > Have the authors considered testing whether other task-specific goals (e.g., skimming) could also be accurately predicted using these models?
> >
>
> In this work we focused on the information-seeking task, which in practice consists of hundreds of different text-specific tasks. In this framework, Shubi and Berzak (2023) have shown that skimming-like behaviour is exhibited in parts of the text. We expect that if skimming is done when reading all the text the task may be even easier. It would be interesting to test out the models on other task-specific goals, however the main bottleneck we encountered is a lack of large scale eye tracking datasets with controlled manipulations that address these tasks (in particular there is currently no available data on skimming).
>
> We addressed your additional comments as follows:
>
> * Complexity in Model Descriptions
>     * Multimodal integration approaches
>
> > Some model descriptions lack clarity, especially in multimodal integration approaches. Providing visual diagrams or more intuitive breakdowns could make these sections more understandable.
> >
>
>  Following your suggestion, we have added more detailed information on the models and included detailed diagrams for all the models in Appendix A.
>
> * Limited Scope of Goal Types:
>
> > The study only explores two reading goals (information seeking and ordinary reading). Extending this research to other goals (e.g., skimming, proofreading) could make the findings more broadly applicable.
> >
>
> Thank you for this suggestion. We refer to this in detail in the response to your second question above. Although we are very interested in these tasks as well, there is currently no available data for them. We intend to collect such data in the future.

---

> > ### Comment · Reviewer_RdFV · 2024-12-02
> >
> > Thank you for the clarifications!

---

### Official Review · Reviewer_yvGR · 2024-11-06

**Soundness:** 3
**Presentation:** 4
**Contribution:** 2
**Rating:** 8
**Confidence:** 5

**Summary:**

In their paper, the authors explore the option of predicting the goal of a person while reading a text. They distuingish between two different natural reading options. The first setting is a ordinary, general reading for pleasure, context, or prose. The second setting is a sharp turn, where the reader has a specfic goal in mind, answering in question, mimicing application scenarios for every person in daily life.

These are the two labels, for several binary classification models that try to distuingish, given the eye movements, and optionally the text, the goal of the reader. They also introduce a new way of interpreting the model's results using linear mixed models.

**Strengths:**

The paper was a pleasure to read. It introduces all relevant background regarding eye movements and reading, has a clear outline and follows a nice story.

Related work is exhaustive and paints a good picture of both machine learning models used in any kind of reading setting as well as eye-tracking-while-reading in general. They then introduce several machine learning models which consume either eye movements, the text that was read during the recording, or both.

**Weaknesses:**

The only true weakness is that the authors do not introduce a new model which exploits the eye movement while reading setting the authors investigate.

Standard error not report, additionally, no statistical significance tests were done between trainable models, e.g. best model vs rest. The critical span (8./9. in the linear mixed effect model) for interpretability only works for already known texts. For binary classification AUROC would also be a great metric to report.

Unfortunately, the anonymous repository is not accessible, as well as the data.

**Questions:**

- Why didn't you report AUROC?

---

> ### Author Response · Authors · 2024-11-19
> **Reply to Reviewer yvGR**
>
> Thank you very much for the in-depth review and the encouraging feedback!
>
> Regarding your question about reporting AUROC, as we mentioned in lines 332-333, we report the ROC curve and AUROC in Appendix D figure 6. Following your comment we further clarified the measures that we report.
>
> Please find responses to your comments and subsequent edits to the paper below.
>
> > Standard error not report
> >
>
> Thank you for this suggestion. We have added standard errors to tables 1 and 3.
>
> > no statistical significance tests were done between trainable models, e.g. best model vs rest.
> >
>
> We have added a statistical significance test between the best model for evaluation each regime, compared to the rest, in both tables 1 and 3.
>
> > The critical span (8./9. in the linear mixed effect model) for interpretability only works for already known texts. For binary classification AUROC would also be a great metric to report.
> >
>
> Thank you for the comment. We acknowledge that in general, annotations of critical spans will not be available. In the dataset used here such annotations do exist and we leverage them for interpretability analysis. However, we note that our models do not assume nor use this information for the prediction task.
>
> > Unfortunately, the anonymous repository is not accessible, as well as the data.
> >
>
> We updated the link, and the current link to the anonymous repository (https://anonymous.4open.science/r/Decoding-Reading-Goals-from-Eye-Movements/ ) should work now. Please let us know if there are any other issues accessing it. The data will be made public as well.

---

> > ### Comment · Reviewer_yvGR · 2024-11-19
> >
> > The improvements look great, thank you for adding it. The link to the repository is working.

---

> > > ### Author Response · Authors · 2024-11-25
> > >
> > > Thank you very much!

---

### Author Response · Authors · 2024-11-19
**Reply to reviewers - summary**

We thank all the reviewers for their very thoughtful reviews and constructive feedback. All four reviewers have a positive opinion of the paper, and found the technical work to be comprehensive and insightful (e.g. Y9QK “experiments are extensive, sound, and deliver some meaningful insights”, v1ny “The methods used are appropriate and the analysis performed is appropriate, well presented, and interpreted”), the research questions and task to be interesting (e.g. v1ny ”the paper proposes an interesting research question”, RdFV “The study introduces a novel problem”) and the paper to be well written (e.g. yvGR “The paper was a pleasure to read”).

We have taken all the comments into careful account, and have made multiple revisions which, as acknowledged by the reviewers, further improved the manuscript. We provide a summary of key revisions here:

- **Additional prior work**: We added all suggested references (as well as several additional references) to the manuscript and revised the related work section. It now better positions our contribution with respect to prior work on related tasks that involve eye movements, and further underlines the novelty of our open-ended question answering setup which enables studying for the first time arbitrary reader goals with respect the text. (lines 492-493, 512-515).
- **Additional statistical testing**: We added statistical tests for best model vs rest, as well as standard errors for all the models in Tables 1 and 3.
- **Improved clarity and detail in model and data exposition**: We made the model descriptions more detailed (Section 3.2) and added diagrams for all the considered models in Figures 5 and 6 in Appendix A. We also added more information on the data: variability of paragraph lengths (lines 216, 230, 233-234) and examples of eyetracking recordings in both conditions in Figure 7 in Appendix B.
- **Expanded discussion**: we expanded the discussion regarding the scientific value and the limitations of the current work (lines 526-533).

Regarding the scope of the paper’s contributions and the fit to ICLR, we would like to stress that first, we define a **new prediction task** of theoretical and practical importance, and benchmark it against state-of-the-art multimodal models for representing eye movements and for integrating them with text. Second, we propose a **new technique for model and task interpretability** by which we show how models can be leveraged for characterizing critical features for the task at hand via mixed effects modeling, a technique that to our knowledge has not been previously used for this purpose and can be applied to analysis of other classification tasks. Finally, we analyze the models’ performance and provide new insights for the new task, which lead to a **new and effective model ensemble** which outperforms all prior methods. This work is therefore highly appropriate for the *applications to cognitive science track* at ICLR, with additional contributions relevant to representation learning for eye movements, *interpretation of learned representations*, and *benchmarks*. More broadly, we believe that the challenge of multimodal modeling and analysis of eye movements and text, which (differently from other types of multimodal data) has not received much attention in ML thus far, will be of high interest to the ICLR community.

Please find our responses to each review below.

---

### Meta-Review · Area_Chair_VyW2 · 2024-12-17

**Metareview:**

This paper initially had mixed reviews. The major issues raised were:

1. no new model is introduced that exploits the eye movements while reading  [yvGR]
2. standard errors not reported [yvGR]
3. extension of previous work, limited novelty [v1ny, Y9QK]
4. lack of discussion of limitations and scientific implications [v1ny]
5. contributions are not strong and original enough to be published at ICLR. In addition, the topic would probably fit better a psycholinguistics or computational linguistics venue. [v1ny]
6. I would not see it at ICLR due to the limited contribution w.r.t. to deep learning, but rather at a more HCI-focused venue such as ETRA, ICMI, IUI,.. [Y9QK]

The authors wrote a response. The response and discussion did not assuage the major concerns of the reviewers. In particular, the major concern is about the novelty (it follows previous studies), and the lack of technical contribution in ML. For the latter, authors claim to have several ML contributions: new technique for model and task interpretability (but this based on existing mixed effects models, L287); new type of model ensemble (but their proposed method is just "stacked generalization" in ML). Thus, the paper lacks ML novelty, and it is not suitable for an "applications track" paper in an ML conference -- The AC agrees with v1ny and Y9QK that the paper is more suitable to an HCI / psycholinguistics / computational linguistics venue.

Authors are encouraged to submit their paper to more suitable venues.

**Additional Comments On Reviewer Discussion:**

see above

---

### Decision · Program_Chairs · 2025-01-22

Reject